# Effects of preoperative stress, depression, social support, and self-compassion on resilience in surgical patients

**Heejoung Lee, Hana Ko** *

College of Nursing, Gachon University, Incheon, South Korea

* hanago11@gachon.ac.kr

## Abstract

### Background

Limited research has been conducted on the psychological factors experienced by patients before surgery. This study examined the effects of preoperative stress, depression, social support, and self-compassion on the resilience of surgical patients.

### Methods

Using convenience sampling, 150 patients admitted for surgery at Nasaret International and Seoul Barun Chuckdo Hospitals in I City, South Korea were included in the study. Data was collected from July 1–31, 2022. Using SPSS WIN 26.0, the data was analyzed using descriptive statistics, a t-test, ANOVA, *Scheffé's* test, Pearson's correlation coefficient, and multiple regression.

### Results

Resilience exhibited a statistically significant negative correlation with surgical stress and depression, and a significant positive correlation with social support. Surgical stress, depression, and obstetrics and gynecological surgery had statistically significant negative effects on resilience, while education level and economic status had positive effects.

### Conclusion

Enhancing resilience in surgical patients requires the development of tailored interventions to mitigate surgical stress and depression, which are patient-centered and account for educational and economic status, particularly in obstetric and gynecological surgery.

**Data availability statement:** All relevant data are within the manuscript files.

**Funding:** The author(s) received no specific funding for this work.

**Competing interests:** The authors have declared that no competing interests exist.

## Introduction

With increased public interest in health and advancements in medical technology and surgical techniques, the number of patients visiting medical institutions is increasing. In 2021, the number of adult surgical patients in South Korea was 1,493,000, accounting for 24% of adult inpatients (6,176,000) and 97% of all surgical patients (1,534,000), and it has continued to increase since then [1].

Patients scheduled for surgery experience various negative emotional reactions, such as activity restrictions, socioeconomic burden, apprehension regarding postoperative sequelae, and deterioration in physical function [2,3]. Undergoing a surgical procedure, even a minor one, is perceived as a major event that cause discomfort and anxiety [3,4]. Such negative emotional reactions can impact the course of surgery and recovery [4].

Resilience is defined as the capacity to overcome stress or adverse experiences [5]. A meta-analysis suggested that higher levels of resilience are associated with faster and more effective recovery from stressful experiences, protection of mental health, and mitigation of the negative effects of a crisis [6], all of which positively affect physical recovery after surgery [7]. In addition, resilience may protect patients with cancer from emotional distress [8] and have a positive effect on self-care behavior [9]. Resilience can be an important factor in overcoming stressful events of surgery as it affects the physical health and contributes to maintaining better health. Therefore, in surgical nursing, it is important to identify and enhance the factors that influence resilience.

In a meta-analysis of resilience, stress was found to be negatively correlated [10]. Stress poses a threat to physical well-being; moreover, surgery is accompanied by physical discomfort and psychological burden [4]. Patients undergoing surgery often experience intense anxiety and depression due to uncertainty about the procedure, fear of anesthesia-related death, and concerns about postoperative pain or complications, all of which create significant stress for both patients and their families [4]. However, recent studies on surgical stress have primarily focused on specific patient groups or patients with cancer, or have measured surgical stress solely using a stress thermometer (100 mm VAS) [11,12].

Depression has been reported to decrease survival rates [13] and affect resilience in patients with colorectal cancer [11]. Moreover, in patients undergoing orthopedic surgery, who require rehabilitation after surgery, the occurrence of depressive symptoms negatively affects postoperative recovery [14–16]. Nursing interventions that promote resilience by continuously assessing and managing depression should be implemented to improve the prognosis of surgical patients [17].

Social support is provided by family, relatives, neighbors, and friends, upon whom an individual can rely for caregiving and from whom they can derive a sense of value [18]. Higher levels of social support are associated with increased postoperative psychosocial functioning, quality of life, and surgical recovery [11,19] underscoring the significance of social support. Furthermore, social support demonstrates a positive correlation with the resilience of patients who have undergone surgical procedures [7,20].

Self-compassion promotes adaptability and resilience in addressing life challenges and represents a crucial construct in mental health and psychological research [21,22]. High levels of self-compassion are associated with a reduced the risk of psychopathology and improved resilience [23].

As indicated by previous research, the relationship between each variable has been examined; however, there is a paucity of studies that have comprehensively investigated these relationships. Research on preoperative psychological factors in surgical patients remains insufficient.

Consequently, to better understand preoperative stress, depression, social support, and self-compassion on resilience in surgical patients, we aim to:1) determine the degree and differences in general characteristics, preoperative stress, depression, social support, self-compassion, and resilience among surgical patients; 2) examine the correlations between preoperative stress, depression, social support, self-compassion, and resilience in surgical patients; and 3) determine the impact of preoperative stress, depression, social support, and self-compassion on the resilience in surgical patients.

## Methods

### Participants

The participants of this study were patients admitted for surgery to N and S general hospitals located in City I. The selection criteria included: 1) adult surgery patients over 20 years of age; 2) patients who were able to communicate, understand, and respond to the contents of the questionnaire; 3) patients without sensory impairment such as hearing and language impairment; and 4) inpatients scheduled for surgical procedures with anesthesia (general, spinal, and brachial plexus anesthesia). The exclusion criteria were as follows: 1) among inpatients, those who were scheduled for surgery with local anesthesia and no anesthesia, and 2) emergency surgery patients.

The sample size required for multiple regression analysis was calculated using G*Power 3.1 (Heinrich-Heine-University, Düsseldorf, Germany), with a significance level of 0.05, an explanatory power of 0.80, and an intermediate effect size of 0.16 [17]. The required sample size for the regression analysis with 11 independent variables was 110 samples. We analyzed all 150 questionnaires completed by participants who consented to participate in the study.

### Data collection

Data was collected from N and S Hospitals in I City, Korea, between July 1 and 31, 2022. After receiving permission from both hospitals, the researchers visited the patients one day before surgery. Data was collected after the participants were informed of the purpose of the study and provided written informed consent. The participants were asked to complete the questionnaires independently; however, if they encountered difficulties, the data collectors read the questions aloud and marked their responses. The participants required approximately 30 minutes to complete the questionnaires and were subsequently presented a gift.

### Measures

**Sociodemographic characteristics.** General characteristics (e.g., age, sex, marital status, religion, occupation, and economic status) were self-reported. Surgery type was categorized under general (appendectomy and pancreatic resection), obstetric and gynecological (hysterectomy and ovarian removal), orthopedic (tendon repair, open and non-open reduction fixation, and artificial joint surgery), and other surgeries (urethral stone removal, incision and biopsy, pin removal, scrotal hydrocele removal, varicose vein removal, carpal tunnel syndrome surgery, and skin grafting).

**Surgical stress.** Surgical stress was measured using a stress measurement tool [24] to assess the emotional state of patients in the preoperative stage. This tool consists of 28 items: twelve surgery-related physical factors, nine

surgery-related emotional factors, and seven surgery-related environmental factors. Each item was scored on a 5-point Likert scale, with higher scores indicating higher stress levels. The Cronbach's α of this scale at the time of its development was 0.92, and it was 0.96 in this study.

**Depression.** Depression was assessed using the Korean version of the Center for Epidemiological Studies Depression Scale (CES-D scale) [25]. It consists of 4 positive and 16 negative items, totaling 20 items. For each item regarding how a patient felt during the past week, "rarely (less than 1 d)" scored 1 point, "sometimes (2-3 days)" 2 points, "very much (4-5 days)" 3 points, and "mostly (6 -7 days)" 4 points. A higher final score, derived by inversely converting the four positive questions, indicates a greater degree of depression. The Cronbach's α of this scale at the time of its development and in this study was 0.87 and 0.91, respectively.

**Social support.** Social support was measured using the Korean version of the Enhancing Recovery in Coronary Heart Disease (ENRICHD) Social Support Instrument (ESSI) [26]. The ES-SI consists of six items, including emotional, informational, and instrumental support, and responses of "yes" or "no" were assigned to each item. In the case of a "yes" response to each item, 1 point was given, and the total score was classified as "poor (3 points or less)," "normal (4 to 5 points)," and "good (6 or more points)." Higher total scores indicate better social support. The Cronbach's α of this scale was 0.84 and 0.87 at the time of its development and in this study, respectively.

**Self-compassion.** Self-compassion was assessed using the Korean version of the self-compassion scale [27]. The Self-Compassion Scale consists of three opposing pairs and six sub-factors: self-kindness, self-judgment, common humanity, isolation, mindfulness, and over-identification. A total of 26 items are scored on a 5-point scale, with total scores ranging from 26 to 130 points. Higher scores indicate more self-compassionate attitudes. The original and current Cronbach's α value was 0.93 and 0.85, respectively.

**Resilience.** The Korean version of the Connor-Davidson Resilience Scale was used [28] to measure resilience. It comprises five factors: tenacity, continuity/endurance, optimism, support, and spirituality. Participants rate each of 25 items on Likert-type scale (0–4): 0 points for "not at all," 1 point for "disagree," 2 points for "normally," 3 points for "mostly so," and 4 points for "very much so," with total score ranging from 0–100. Higher values indicate greater resilience. The Cronbach's α of this scale at the time of its development was .89 and was .95 in this study.

## Ethical considerations

This current study was reviewed and approved by the Institutional Review Board of Gachon University (approval no.1044396-202206-HR-125-01). Written consent to participate in the study was obtained from the research participants one day before surgery. We informed participants that they could withdraw from the study at any time. To ensure the confidentiality of their personal information, completed questionnaires were placed in sealed, opaque envelopes, and each questionnaire was assigned a unique identification number to prevent identification during data analysis. The questionnaires were stored in a locked, designated box, accessible only to the researcher directly involved in the study. The research database was securely stored on a computer with restricted access to safeguard it from external exposure during document preparation and data management.

## Data analysis

The data collected in this study was analyzed using IBM SPSS version 26.0. All data are presented as descriptive statistics, including means and standard deviations. Differences between the general characteristics and resilience of the participants were analyzed using an independent t-test and one-way analysis of variance (ANOVA), and the *Scheffé* test was used for the post hoc test. Skewness and kurtosis were calculated to determine whether the assumptions of normality of the variables were satisfied. Correlations between preoperative stress, depression, social support, self-compassion, and resilience were analyzed using Pearson's correlation coefficients. Hierarchical multiple linear regression analysis was used to determine whether participants' preoperative stress, depression, social support, and self-compassion affected their resilience.

## Results

### Resilience according to the sociodemographic characteristics of the surgical patients

The patients' sociodemographic data is presented in Table 1. Of the 150 participants, 85 (56.7%) were men, and the mean score for resilience was 89.66 ± 14.31. Thirty-nine patients (26.0%) were in their 40s, and the mean score for resilience was 91.18 ± 12.62. There were statistically significant differences in resilience according to educational level (F = 7.153, $p$ = .001), living arrangements (F = 4.238, $p$ = .016), employment status (t = 2516, $p$ = .016), economic status (F = 8.630, $p$ < .001), and type of surgery (F = 5.366, $p$ = .002).

### Degree of stress, depression, social support, self-compassion, and resilience before surgery

The average score for surgical stress was 59.61 out of 140 points, indicating a moderate level of stress. The sub-factors had average scores of 27.53 points for physical factors (moderate), 17.52 points for emotional factors (low), and 14.56 points for environmental factors (low). The total score for depression was 33.97, indicating moderate depression. The average score social support was 5.47 out of 6, indicating a high support. The total score for self-compassion was 72.62 out of 130, indicating a moderate self-compassion. The sub-factors of self-compassion were common humanity (12.37,

**Table 1. Resilience values of the participants by sociodemographic characteristics (N = 150).**

| Characteristics | Category | N(%) | Resilience values | | | |
| --- | --- | --- | --- | --- | --- | --- |
| | | | Mean(SD) | t or F | p | Scheffé |
| Sex | Male | 85 (56.7) | 89.66 (14.31) | 1.973 | .050 | |
| | Female | 65 (43.3) | 84.85 (15.44) | | | |
| Age | 20's | 27 (18.0) | 90.41 (16.74) | 2.352 | .057 | |
| | 30's | 21 (14.0) | 86.10 (15.35) | | | |
| | 40's | 39 (26.0) | 91.18 (12.62) | | | |
| | 50's | 37 (24.7) | 87.43 (10.86) | | | |
| | 60+ | 26 (17.3) | 80.62 (18.95) | | | |
| Educational level | Junior high school[a] | 19 (12.6) | 77.21 (18.45) | 7.153 | .001 | a<b,c |
| | High school[b] | 59 (39.3) | 86.68 (13.89) | | | |
| | ≥College[c] | 72 (48.0) | 91.04 (13.55) | | | |
| Marital status | Married[a] | 90 (60.0) | 87.93 (12.68) | 4.238 | .016 | c<a,b |
| | Unmarried[b] | 46 (30.7) | 90.04 (16.57) | | | |
| | Other (separation, divorce, bereavement, etc.)[c] | 14 (9.3) | 77.14 (19.24) | | | |
| Religion | Buddhism | 10 (6.7) | 91.10 (16.62) | 2.607 | .054 | |
| | Christian | 30 (20.0) | 81.07 (19.56) | | | |
| | Catholic | 24 (16.0) | 90.50 (17.17) | | | |
| | Other(none) | 86 (57.3) | 88.62 (11.50) | | | |
| Employment status | Employed | 117 (78.0) | 89.51 (13.09) | 2.516 | .016 | |
| | Unemployed | 33 (22.0) | 80.70 (18.89) | | | |
| Economic status | Low[a] | 21 (14.0) | 76.38 (16.07) | 8.630 | <.001 | a<b,c |
| | Middle[b] | 117 (78.0) | 88.80 (13.68) | | | |
| | High[c] | 12 (8.0) | 95.17 (16.34) | | | |
| Type of surgery | General surgery[a] | 26 (17.4) | 85.46 (18.44) | 5.366 | .002 | b<a,c,d |
| | Obstetrics and Gynecology surgery[b] | 12 (8.0) | 72.58 (20.78) | | | |
| | Orthopedic surgery[c] | 61 (40.7) | 89.52 (12.36) | | | |
| | Others(urethral stone removal, incision and biopsy, pin removal, etc.)[d] | 51 (34.0) | 89.84 (12.31) | | | |

moderate), isolation (9.55, low), mindfulness (13.19, moderate), and over-identification (10.09, low). The total resilience score was 87.57 out of 125, indicating a moderate level of resilience (Table 2).

### Correlation between stress, depression, social support, self-compassion, and resilience before surgery

Resilience was significantly negatively correlated with surgical stress (r = -.356, $p < .001$) and depression (r = -.452, $p < .001$) and significantly positively (+) correlated with social support (r = .354, $p < .001$). Self-compassion was significantly positively correlated with surgical stress (r = .293, $p < .001$) and depression (r = .278, $p < .001$) and negatively correlated with social support (r = -.194, $p < .05$). Social support was significantly and negatively correlated with surgical stress (r = -.417, $p < .001$) and depression (r = -.529, $p < .001$). Depression showed a statistically significant positive (+) correlation with surgical stress (r = .655, $p < .001$) (Table 3).

### Factors affecting the resilience of patients before surgery

Hierarchical multiple linear regression analysis was conducted to examine the effects of preoperative stress, depression, social support, and self-compassion on resilience. In the case of self-compassion, no correlation was found with the results of this study. The demographic and sociological background variables that were found to be significantly related to resilience through a difference test, including educational level, marital status, employment status, economic status, and

**Table 2. Degrees of stress, depression, social support, self-compassion, and resilience before surgery (N = 150).**

| Variable | Categories | Range | Mean (SD) | Skewness | Kurtosis |
|---|---|---|---|---|---|
| Surgical stress | Physical component | 12–54 | 27.53 (9.68) | 0.54 | -0.46 |
| | Emotional component | 8–38 | 17.52 (7.15) | 0.77 | 0.10 |
| | Environmental factors | 7–33 | 14.56 (5.39) | 0.83 | 0.71 |
| | Total score | 28–124 | 59.61 (20.91) | 0.69 | 0.00 |
| Depression | Total score | 20–69 | 33.97 (8.84) | 0.99 | 1.00 |
| Social support | Total score | 0–6 | 5.47 (1.34) | −2.89 | 7.96 |
| Self-compassion | Self-kindness | 7–25 | 15.42 (2.81) | −0.01 | 1.24 |
| | Self-judgment | 6–25 | 12.00 (3.16) | 0.69 | 1.33 |
| | Common humanity | 4–20 | 12.37 (2.83) | −0.16 | 0.49 |
| | Isolation | 4–20 | 9.55 (2.64) | 0.54 | 1.00 |
| | Mindfulness | 4–20 | 13.19 (2.55) | −0.57 | 0.90 |
| | Over-identification | 4–20 | 10.09 (2.69) | 0.43 | 0.33 |
| | Total score | 36–130 | 72.62 (10.83) | 0.40 | 5.36 |
| Resilience | Total score | 25–125 | 87.57 (14.95) | −0.38 | 1.80 |

**Table 3. Correlations among preoperative stress, depression, social support, self-compassion, and resilience (N = 150).**

| | Surgical stress | Depression | Social support | Self-compassion | Resilience |
|---|---|---|---|---|---|
| Surgical stress | 1 | | | | |
| Depression | .655** | 1 | | | |
| Social support | −.417** | −.529** | 1 | | |
| Self-compassion | .293** | .278** | −.194* | 1 | |
| Resilience | −.356** | −.452** | .354** | −.140 | 1 |

*$p < .05$

** $p < .001$

surgery type, were controlled for. The VIF index was less than 10 (1.203–2.225), which did not cause multicollinearity, and the Durbin-Watson coefficient was 1.958, which was close to 2, thus satisfying the assumption of residual independence.

In Model 1, control variables such as educational level, marital status, employment status, economic status, and surgery type were treated as dummy variables and inputs. In Model 2, surgical stress, depression, social support, and self-compassion were used as inputs to examine their effects on resilience.

First, in Model 1, in which the control variables were input, economic status ($β$=.242, $p<.01$) and gynecological surgery ($β$=-.251, $p<.01$) were found to have a significant effect on resilience and appeared to have an explanatory power of 18.9% (F = 5.350, $p<.001$).

As a result of the additional inputs of surgical stress, depression, social support, and self-compassion in Model 2, the explanatory power increased by 14.7% compared to Model 1, resulting in a total explanatory power of 32.6% (F = 6.996, $p<.001$). Surgical stress ($β$=-.185, $p<.05$) and depression ($β$=-.204, $p<.05$) had significant negative effects on resilience, whereas educational level ($β$=.183, $p<.05$), economic status ($β$=.191, $p<.05$), and gynecological surgery ($β$=-.173, $p<.05$) affected resilience (Table 4).

## Discussion

This descriptive study analyzed the effects of preoperative stress, depression, social support, and self-compassion on resilience in surgical patients to provide evidence for comprehensive and personalized nursing support to aid rapid recovery after surgery.

The mean preoperative stress score was 59.61, and the mean depression score was 33.97, supporting the findings of a previous study [24] that surgical patients experienced moderate stress and depression. Patients who are about to undergo surgery, regardless of the extent of the surgery, experience high levels of stress and depression owing to anxiety about the surgery and concerns regarding the outcomes.

We also observed that patients who underwent surgery had a high social support score of 5.47. This finding may be attributed to the fact that social support increases when individuals experience stressful situations, during illness or crises,

Table 4. Effects of preoperative stress, depression, social support, and self-compassion on resilience (N = 150).

| Predictors | Model 1 | | | | Model 2 | | | |
|---|---|---|---|---|---|---|---|---|
| | B | β | t | p | B | β | t | p |
| Educational level | 2.707 | .126 | 1.404 | .163 | 3.938 | .183 | 2.162 | .032 |
| Married | −1.828 | −.060 | −0.725 | .470 | −0.387 | −.013 | −0.166 | .868 |
| Employed | 0.976 | .027 | 0.309 | .758 | −0.682 | −.019 | −0.235 | .815 |
| Economic status | 7.748 | .242 | 2.990** | .003 | 6.130 | .191 | 2.560 | .012 |
| General surgery | −5.166 | −.131 | −1.587 | .115 | −3.176 | −.081 | −1.051 | .295 |
| Obstetrics and Gynecology surgery | −13.779 | −.251 | −3.061 | .003 | −9.524 | −.173 | −2.268 | .025 |
| Orthopedic surgery | 0.833 | .027 | 0.323 | .747 | 2.275 | .075 | 0.948 | .345 |
| Surgical stress | | | | | −0.132 | −.185 | −1.987* | .049 |
| Depression | | | | | −0.344 | −.204 | −2.030* | .044 |
| Social support | | | | | 0.978 | .088 | 1.071 | .286 |
| Self-compassion | | | | | 0.024 | .017 | 0.230 | .818 |
| R² (adjR²) | .233 (.189) | | | | .380 (.326) | | | |
| F | 5.350 | | | | 6.996 | | | |
| P | <.001 | | | | <.001 | | | |

Note: Marital status (not married = 0), Employment (unemployed = 0), Surgery (Others = 0)

or those involving conflict, frustration, or changes requiring adjustment [29]. Therefore, interventions are needed to assist individuals with high levels of social support in buffering against stressful events.

After identifying the factors influencing patients' resilience before surgery, the explanatory power was found to be 32.6%. Factors such as degree of education, economic status, surgery type (obstetrics and gynecology), surgical stress, and depression had a significant effect on resilience. The results of this study are consistent with those of other studies [11], which reported that the site of surgery impacts resilience and that the presence or absence of a stoma after colorectal cancer surgery also affects resilience. The findings of this study indicate that the type of surgery, even if considered uncomplicated, can significantly affect patient resilience. In addition, gynecological surgery patients showed lower resilience than other patients [30]. Obstetrics and gynecology surgeries are often viewed as procedures involving organs that represent key aspects of womanhood, such as maternal love, sexual attraction, and femininity [17,31]. These results underscore the importance of implementing interventions to enhance resilience, particularly for patients undergoing gynecological surgery.

Economic status also appears to affect resilience. This finding suggests that economic conditions can increase resilience [11]. In Korea, patients with cancer receive significant support from the government through special evaluations. However, the participants in the present study were in the high-risk group and did not receive this support. Therefore, the cost burden rate is high. Hence, there is a need to increase the resilience of these individuals through economic support. Recently, the expansion of recipients of the next-highest level of out-of-pocket burden reduction support for a group with a relatively large economic burden of medical expenses was also suggested as a support plan for the medically vulnerable group [32].

The finding that depression is an influencing factor of resilience is consistent with the results of previous studies [2,10]; lower levels of depression before surgery indicate higher patient resilience. Depression can be reduced through frequent visits by the family; increased level of family support appears to lower the degree of depression. Visits can serve as social support, helping patients reduce their degree of depression. In addition, video calls from family members and acquaintances were effective in reducing depression among older residents of nursing facilities who were not allowed visitors owing to COVID-19 [33]. Visiting is an action that goes beyond a simple meeting, as it reflects support and consideration, making it an effective way to alleviate patients' depression. Providing preoperative education and preparation of patients and their families as along with educating them on post-discharge patient management, will reduce uncertainty regarding the surgical process and alleviate depression by fostering a sense of security. Moreover, family support is crucial for obstetric patients, as those undergoing obstetric surgery may experience negative self-perceptions and fear of the procedure. Encouraging positive thinking has been shown to reduce depression [34], highlighting the need for emotional support interventions that promote positive thinking among perioperative patients.

Additionally, lower levels of surgical stress indicate a higher resilience. A review of the relationship between distress and resilience in patients with brain injury and cancer [35,36] found a negative impact; the observation was consistent with the results of our study. Resilience-promoting interventions have been reported to improve positive adaptation and quality of life in patients with cancer [37]; therefore, it may be valuable to develop and evaluate interventions for surgical patients in the future. For example, physical activity helps reduce stress, improve physical and mental health, and enhance the quality of life [38–40].

Furthermore, the relationship between self-compassion and stress varies across studies. In this study, self-compassion was found to be positively correlated with depression and surgical stress and negatively correlated with social support; however, it did not impact resilience. Self-compassion was inversely correlated with isolation and depression among college students during the COVID-19 pandemic [41]. In addition, studies on female adolescents have shown that self-compassion reduces negative emotions, such as depression and anxiety [21,42]. However, previous studies examining different subsets of self-compassion have shown individuals with high levels of self-criticism appear to focus more on their mistakes or failure-related stimuli, which increases their stress and makes them more susceptible to depression [43]. Increasing self-compassion and reducing self-criticism have been suggested as effective ways to overcome these

issues. Although this study did not analyze self-criticism and self-compassion separately, patients undergoing surgery are reported to be in stressful situations revealing high levels of self-criticism [30]. Therefore, the correlation between self-compassion and negative emotions such as stress and depression in patients' preoperative needs to be investigated by subcategorizing self-compassion into self-compassion and self-criticism through rigorous studies. This observation suggests that although support from family and friends is important before surgery, preventing excessive emotions, such as self-compassion and self-criticism, in patients is vital in increasing resilience.

Several limitations of this study should be considered when interpreting the results. First, as this study only selected hospital patients who visited rural areas rather than large cities, the sample may be biased. Therefore, the results of this study cannot be generalized to surgical patients in the entire community. Second, as most of the recruited participants underwent low-risk surgeries such as appendectomy, hysterectomy, ovarian removal, tendon repair, and open and closed reduction fixation, caution should be exercised when applying the results of this study to high-risk surgeries such as cancer, neurological, and cardiac surgeries. Third, the study design was cross-sectional, indicating that the data were collected at a single point in time. This factor limits their ability to identify changes in assessments over time. Longitudinal studies are beneficial for understanding stress and resilience in patients undergoing surgery. Finally, resilience is inherently subjective, and its assessment can be influenced by personal and cultural norms. Although some instruments have been validated, individual interpretations may vary. Future research is needed to test reliability and validity in surgical patients, identification of sub-factors through factor analysis, and evaluation of socio-cultural factors from multiple perspectives.

## Conclusions

Higher education level, higher economic status, lower surgical stress, depression, and gynecologic surgery were found to influence surgical patients' resilience. These findings suggest that multifaceted aspects should be considered when developing interventions to improve the resilience of patients undergoing surgery. Efficient intervention programs may be developed to reduce preoperative stress and depression, particularly in obstetric and gynecological surgery. Furthermore, economic support for surgery should be expanded for vulnerable groups.

## Acknowledgments

The authors thank the participants for making this study possible.

## Author contributions

**Conceptualization:** Heejoung Lee, Hana Ko.

**Data curation:** Heejoung Lee.

**Methodology:** Heejoung Lee, Hana Ko.

**Software:** Heejoung Lee.

**Supervision:** Hana Ko.

**Validation:** Hana Ko.

**Visualization:** Hana Ko.

**Writing – original draft:** Heejoung Lee, Hana Ko.

**Writing – review & editing:** Heejoung Lee, Hana Ko.

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
