## [Decision Letter · Decision Letter 0]

28 Nov 2024

PONE-D-24-41400Effects of preoperative stress, depression, social support, and self-compassion on the resilience in surgical patientsPLOS ONE

Dear Dr. Ko,

Thank you for submitting your manuscript to PLOS ONE. After careful consideration, we feel that it has merit but does not fully meet PLOS ONE’s publication criteria as it currently stands. Therefore, we invite you to submit a revised version of the manuscript that addresses the points raised during the review process.

We look forward to receiving your revised manuscript.

Kind regards,

Kamalakar Surineni, MD, MPH

Guest Editor

PLOS ONE

Journal Requirements:

Reviewers' comments:

Reviewer's Responses to Questions

**Comments to the Author**

1. Is the manuscript technically sound, and do the data support the conclusions?

Reviewer #1: Yes

Reviewer #2: Yes

2. Has the statistical analysis been performed appropriately and rigorously? 

Reviewer #1: Yes

Reviewer #2: Yes

3. Have the authors made all data underlying the findings in their manuscript fully available?

Reviewer #1: Yes

Reviewer #2: Yes

4. Is the manuscript presented in an intelligible fashion and written in standard English?

Reviewer #1: Yes

Reviewer #2: Yes

5. Review Comments to the Author

Reviewer #1: The study titled "Effects of Preoperative Stress, Depression, Social Support, and Self-Compassion on the Resilience in Surgical Patients" presents a valuable exploration into psychological factors that influence resilience before surgery. I would suggest these changes/limitations to be added:

1. The exclusion criteria mentions psychiatric diagnosis such as depression and schizophrenia due to possible reasons of being unable to understand the questionnaire. However it is unclear if any cognitive assessment was done to arrive at this decision. If not, it would be presumptive to assume that psychiatric diagnoses is associated with cognitive impairment which is always not the case. This also limits the ability to generalize findings.

2. Collecting data one day before surgery may not capture the psychological fluctuations that could occur closer to surgery (e.g., weeks before surgery, immediately before, and post-surgery). This does not allow to observe changes in stress and resilience over time.

3. Resilience is heavily influenced by cultural contexts. While the tools used are validated, some of the scales (e.g., self-compassion) may not capture all relevant cultural or context-specific nuances. This limitation needs to be added.

4. The study found no significant correlation between self-compassion and resilience, contrary to expectations based on previous literature. It would be beneficial to explore why this discrepancy occurred, possibly through a deeper analysis of the cultural or contextual factors influencing self-compassion in the study population.

Reviewer #2: Overall, the authors have done a good job in discussing the need for a new research study and

Abstract: Overall discusses the topic appropriately. However, I would avoid use of embellishments such as the word “basic data” as there is more complexity than that word implies and diminishes the importance of the data. Also, “quick recovery” is mentioned in abstract but is not quite mentioned in the body of the article, so it does not seem to fit. Other grammatical errors, such as past tense should be “data was collected” rather than “data were collected” at various places in the abstract. Also mention how OB surgery affected resilience as results should be included, as current sentence about it seems ambiguous. Another thing that could be considered is putting 4 sections in the abstract, such as background, methods, results, and conclusion to make the abstract more streamlined. Many papers have these sections in abstract to provide more clarity. This is not necessary but can be considered by authors.

Introduction: There is good information in this section about prior research studies on this topic and the need for a new research question. There are many grammatical errors in the writing, however, that need revision. For example, the first sentence in this section is not a complete sentence. On page 3, the sentence “The meta-analysis suggested that…” is a run-on sentence and grammar needs revision. Same with next sentence “In addition, resilience…” In the prior paragraph, “emotional confusion” is mentioned but it is not clear what this means and how it’s supported by research. In the last para, it mentions “Patients undergoing surgery also experience severe pressure and depression…” The word “severe” seems like an embellishment and absolute to all patients undergoing surgery, which is not clearly shown how it is supported by research. The next sentence starts with “However, recent studies of surgical stress…” The “however” word does not seem to fit as it does not seem to relate clearly in that way to the previous sentence. There are other overall grammatical errors in the introduction that requires more work.

Methods:

The methods are described reasonably well. It is well-presented with different headings and each parameter is discussed separately (such as self-compassion), which is great. There are again English errors that need correction. On page 6, first paragraph again needs major revisions in the English sentence structure and grammar. There is a flipping between past and present tense as well, which would need change and kept consistent instead. On page 7, the last line is “To measure self-compassion…” This is not a complete sentence.

Results:

Overall well presented with good use of tables. Sentence structure again needs work at some places. For example, on page 9 it states “The 39 patients were in 40s…” does not require “The” word and can be presented better. On page 11, it would be helpful for authors to describe what these values mean subjectively when interpreted. For example, was the total score of surgical stress represent overall moderate level of stress, or low or high, etc.? This would make the article easier to follow for the reader. On page 12, it states “self-compassion was significantly positively correlated with surgical stress…”. This suggests that higher self-compassion means more surgical stress, which is counter-intuitive. Please clarify potential reasons for this, or correct if it was an error. Later in that page, there is a mention of “self-regret” which is not explained anywhere in the article. Only self-compassion was explained. Later it also mentions “self-disappointment”. Authors should avoid introducing new variables that have not been explained earlier.

Discussion: Overall this was a section that discussed the results and limitations well. Again, it mentions self-regret which is not clarified or explained earlier in the article. On page 14, the sentence “We also found that patients who underwent surgery showed high social support was.” Is not a correct sentence. On page 15, it mentions colorectal cancer and it is being made parallel to gynecological surgery, when they are very different. Later there is a mention of “special calculations” and unclear what this means. On page 16, the sentence “However, prior education on surgery methods…” is a run-on sentence. There are grammatical errors on this page. The discussion also does not include self-compassion which was one key parameter.

Conclusions: It is reasonably concise.

Overall, authors do present a good research question and it seems to be a good study design. However, there are many grammatical/English errors that require revision to improve clarity of the article.

6. PLOS authors have the option to publish the peer review history of their article (what does this mean? ). If published, this will include your full peer review and any attached files.

**Do you want your identity to be public for this peer review?** For information about this choice, including consent withdrawal, please see our Privacy Policy .

Reviewer #1: **Yes: ** Nikhil Tondehal

Reviewer #2: No

---

## [Author Response · Author response to Decision Letter 1]

19 Dec 2024

Dear Editor and Reviewers:

We would like to thank you for your thoughtful comments and recommendations. We have made the following changes to the manuscript to address your concerns and revised the paper, paying particular attention to language. The revised sections of the manuscript are indicated in RED in the text.

Please see the revised manuscript and response to reviewers files.

---

## [Decision Letter · Decision Letter 1]

5 Feb 2025

PONE-D-24-41400R1Effects of preoperative stress, depression, social support, and self-compassion on resilience in surgical patientsPLOS ONE

Dear Dr. Ko,

Thank you for submitting your manuscript to PLOS ONE. After careful consideration, we feel that it has merit but does not fully meet PLOS ONE’s publication criteria as it currently stands. Therefore, we invite you to submit a revised version of the manuscript that addresses the points raised during the review process.

We look forward to receiving your revised manuscript.

Kind regards,

Kamalakar Surineni, MD, MPH

Guest Editor

PLOS ONE

Journal Requirements:

Reviewers' comments:

Reviewer's Responses to Questions

**Comments to the Author**

1. If the authors have adequately addressed your comments raised in a previous round of review and you feel that this manuscript is now acceptable for publication, you may indicate that here to bypass the “Comments to the Author” section, enter your conflict of interest statement in the “Confidential to Editor” section, and submit your "Accept" recommendation.

Reviewer #1: All comments have been addressed

Reviewer #2: (No Response)

Reviewer #3: (No Response)

2. Is the manuscript technically sound, and do the data support the conclusions?

Reviewer #1: Yes

Reviewer #2: Yes

Reviewer #3: Yes

3. Has the statistical analysis been performed appropriately and rigorously? 

Reviewer #1: Yes

Reviewer #2: I Don't Know

Reviewer #3: Yes

4. Have the authors made all data underlying the findings in their manuscript fully available?

Reviewer #1: Yes

Reviewer #2: Yes

Reviewer #3: Yes

5. Is the manuscript presented in an intelligible fashion and written in standard English?

Reviewer #1: Yes

Reviewer #2: No

Reviewer #3: Yes

6. Review Comments to the Author

Reviewer #1: (No Response)

Reviewer #2: Abstract: Authors have done a good job with dissecting the abstract into 4 separate sections. In Methods, it should say “Data was collected…” and “data was analyzed…” rather than “were”. In Results part of the abstract, it should also be clarified if the data on higher education, economic status, and gynecological surgery suggested a positive or negative correlation to resilience and if it was statistically significant or not.

Introduction: There are multiple grammatical errors in this section. For example, in the first para, it states that in 2021, number of patients undergoing surgical treatment will increase in 2021. However, we are already in 2025 so this statement is not accurately worded. In the second para, it states “…is perceived as major events…” but it should be “…is perceived as a major event…”.

In the third paragraph, it states: “Resilience can be an important factor in overcoming stressful events of surgery as it affects the physical and contributes to maintaining better health. Therefore, in surgical nursing, it is important to identify and enhance the factors that influence resilience.” This section can be better worded as it appears very generic rather than worded for a study.

On page 4, it states “patients undergoing surgery often experience intense pressure and depression…”. Please clarify what do we mean by “intense pressure” which sounds quite ambiguous? Or do we mean stress? Also, later in that paragraph, the word “nevertheless” appears out of place.

In another part, the sentence should be modified to: “Moreover, in patients undergoing orthopedic surgery…”.

The last paragraph of the introduction appears very convoluted and hard to read. It could be better worded for the reader.

Methods:

In data collection, it should read: “Data was collected after the participants…”. In Data analysis, it should read: “The data collected in this study was analyzed using IBM…” and there are other portions of the paragraph that requires the same correction in tense.

Results: The first line should read: “The patients’ sociodemographic data is presented…” Later, on page 11, it states “the total score for depression was 33.97 80” which looks like an error. On page 13, it states “However, based on previous studies, it was included and its effect on resilience was examined”. This sentence seems out of place for Results section, as it should only discuss results of the current study. Discussion about other studies can be provided in the discussion section instead.

Discussion: I would recommend authors to avoid words such as “basic data” as it seems out of place in a study that is analyzing various parameters. On page 16, it states “visiting is an acting that goes beyond a simple meeting…” which can be omitted or better worded. Overall, this section provides a good discussion, including one about limitations of the study. Authors have overall done a good job in this section. However, the language / wording could be streamlined at places.

Overall, authors have presented a good research question and methods for a study. Authors can revise the language further as there are several grammatical and other typos that need corrected.

Reviewer #3: (No Response)

7. PLOS authors have the option to publish the peer review history of their article (what does this mean? ). If published, this will include your full peer review and any attached files.

**Do you want your identity to be public for this peer review?** For information about this choice, including consent withdrawal, please see our Privacy Policy .

Reviewer #1: **Yes: ** Nikhil Tondehal

Reviewer #2: No

Reviewer #3: **Yes: ** Anoop Narahari

---

## [Author Response · Author response to Decision Letter 2]

24 Feb 2025

We would like to thank you for your thoughtful comments and recommendations. We have made the following changes to the manuscript to address your concerns and revised the paper, paying particular attention to language. The revised sections of the manuscript are indicated in RED in the text.

---

## [Decision Letter · Decision Letter 2]

26 Mar 2025

Effects of preoperative stress, depression, social support, and self-compassion on resilience in surgical patients

PONE-D-24-41400R2

Dear Dr. Hana Ko,

We’re pleased to inform you that your manuscript has been judged scientifically suitable for publication and will be formally accepted for publication once it meets all outstanding technical requirements.

Kind regards,

Kamalakar Surineni, MD, MPH

Guest Editor

PLOS ONE

Additional Editor Comments (optional):

Thank you for your efforts in addressing the reviewer feedback and improving the manuscript. It is scientifically sound and meets the publication criteria of the journal. I am happy to inform you that it has been accepted.

Reviewers' comments:

Reviewer's Responses to Questions

**Comments to the Author**

1. If the authors have adequately addressed your comments raised in a previous round of review and you feel that this manuscript is now acceptable for publication, you may indicate that here to bypass the “Comments to the Author” section, enter your conflict of interest statement in the “Confidential to Editor” section, and submit your "Accept" recommendation.

Reviewer #1: All comments have been addressed

Reviewer #3: All comments have been addressed

2. Is the manuscript technically sound, and do the data support the conclusions?

Reviewer #1: Yes

Reviewer #3: Yes

3. Has the statistical analysis been performed appropriately and rigorously? 

Reviewer #1: I Don't Know

Reviewer #3: Yes

4. Have the authors made all data underlying the findings in their manuscript fully available?

Reviewer #1: Yes

Reviewer #3: Yes

5. Is the manuscript presented in an intelligible fashion and written in standard English?

Reviewer #1: Yes

Reviewer #3: Yes

6. Review Comments to the Author

Reviewer #1: (No Response)

Reviewer #3: (No Response)

7. PLOS authors have the option to publish the peer review history of their article (what does this mean? ). If published, this will include your full peer review and any attached files.

**Do you want your identity to be public for this peer review?** For information about this choice, including consent withdrawal, please see our Privacy Policy .

Reviewer #1: **Yes: ** Nikhil Tondehal

Reviewer #3: **Yes: ** Anoop Narahari

---

## [Editor Report · Acceptance letter]

PONE-D-24-41400R2

PLOS ONE

Dear Dr. Ko,

I'm pleased to inform you that your manuscript has been deemed suitable for publication in PLOS ONE. Congratulations! Your manuscript is now being handed over to our production team.

Kind regards,

on behalf of

Dr. Kamalakar Surineni

Guest Editor

PLOS ONE